# Blood Levels of Endocannabinoids, Oxylipins, and Metabolites Are Altered in Hemodialysis Patients

**DOI:** 10.3390/ijms23179781

**Published:** 2022-08-29

**Authors:** Bruce A. Watkins, Allon N. Friedman, Jeffrey Kim, Kamil Borkowski, Shaun Kaiser, Oliver Fiehn, John W. Newman

**Affiliations:** 1Department of Nutrition, University of California, Davis, CA 95616, USA; 2University Hospital, Suite 6100, Indiana University School of Medicine, Indianapolis, IN 46202, USA; 3Department of Internal Medicine, University of California, Davis, CA 95616, USA; 4West Coast Metabolomics Center, Genome Center, University of California, Davis, CA 95616, USA; 5Texas Kidney Institute, Dallas, TX 75230, USA; 6West Coast Metabolomics Center, University of California, Davis, CA 95616, USA; 7Obesity and Metabolism Research Unit, USDA-ARS Agriculture Research Service, Davis, CA 95616, USA

**Keywords:** hemodialysis patient, women, endocannabinoids, oxylipins, global metabolites, polyunsaturated fatty acids

## Abstract

Hemodialysis patients (HDPs) have higher blood pressure, higher levels of inflammation, a higher risk of cardiovascular disease, and unusually low plasma n-3 polyunsaturated fatty acid (PUFA) levels compared to healthy subjects. The objective of our investigation was to examine the levels of endocannabinoids (eCBs) and oxylipins (OxLs) in female HDPs compared to healthy matched female controls, with the underlying hypothesis that differences in specific PUFA levels in hemodialysis patients would result in changes in eCBs and OxLs. Plasma phospholipid fatty acids were analyzed by gas chromatography. Plasma was extracted and analyzed using ultra-performance liquid chromatography followed by electrospray ionization and tandem MS for eCBs and OxLs. The global untargeted metabolite profiling of plasma was performed by GCTOF MS. Compared to the controls, HDPs showed lower levels of plasma EPA and the associated OxL metabolites 5- and 12-HEPE, 14,15-DiHETE, as well as DHA derived 19(20)-EpDPE. Meanwhile, no changes in arachidonylethanolamide or 2-arachidonylglycerol in the open circulation were detected. Higher levels of multiple N-acylethanolamides, monoacylglycerols, biomarkers of progressive kidney disease, the nitric oxide metabolism-linked citrulline, and the uremic toxins kynurenine and creatine were observed in HDP. These metabolic differences in cCBs and OxLs help explain the severe inflammatory and cardiovascular disease manifested by HDPs, and they should be explored in future studies.

## 1. Introduction

Hemodialysis patients (HDPs) suffer from malnutrition [1,2,3,4,5] and generalized systemic inflammation [4]. US HDPs have a blood distribution of long-chain n–3 PUFAs that is similar to that of the general population, but their n-3 PUFA blood content is among the lowest recorded in the medical literature [1]. Moreover, long-chain n-3 PUFAs are strongly and independently associated with a lower risk of sudden cardiac death in hemodialysis patients throughout the first year of hemodialysis [6]. What is less understood is if changes in endocannabinoids (eCBs) and oxylipins (OxLs) are contributing factors in, or a consequence of, kidney disease. The global analysis of blood metabolites can be useful to ascertain how malnutrition, the loss of kidney function, and inflammation influence macronutrient metabolism in hemodialysis patients [7].

The endocannabinoid system (ECS) functions with its bioactive lipid mediators and receptors to perform a host of behavioral and biochemical functions—for instance, modulating food intake [8], pain perception [9], and memory processes [10]. The ECS ligand arachidonoylethanolamine (anandamide, AEA), which is derived from arachidonic acid (AA), was the first endogenous ligand identified [11], followed by the identification of 2-arachidonoylglycerol (2-AG) [12,13]. Since then, numerous ECS ligands have been isolated and identified [14,15]. While the ECS is known to influence renal physiology [16,17,18], its role in HDPs has not been well described.

Similar to eCBs, OxLs are another group of endogenously produced lipid-derived biologically active signaling compounds that act locally with broad physiological and pathological effects on cellular function, acting through receptor- and ion channel-mediated processes, and that play both pro- and anti-inflammatory roles in a number of pathological processes [19,20,21]. The OxLs encompass a wide range of compounds with diverse chemical structures, such as prostaglandins, leukotrienes, di-and tri-hydroxy pro-resolving lipid mediators (e.g., lipoxins and resolvins), and various epoxides and their diol metabolites formed by the actions of cyclooxygenases, lipoxygenases, and cytochrome P450s and epoxide hydrolases, or through the interaction of PUFAs with reactive oxygen species [19]. Among the diverse functions of OxLs, many of these lipid mediators regulate smooth muscle tone, which affects small blood vessel dilation, influences inflammation, regulates blood pressure, and alters cardiovascular disease risk and kidney function [21,22,23].

The long-chain n-3 PUFA status and the effects of supplementing with fish oil have been extensively studied to show the link between dietary n-3 PUFA intake and their levels in circulation for subjects undergoing hemodialysis. Several investigators [6,24,25] have characterized the circulating levels of n-3 PUFA (eicosapentaenoic acid (EPA) and docosahexaenoic acid (DHA)) in populations of hemodialysis patients and found that EPA and DHA are possible modifiable cardiovascular risk factors of sudden cardiac death. As precursor PUFAs for both eCBs and OxLs, changes in EPA, DHA, and the n-6 PUFA (arachidonic acid (AA)) status in a human would inevitably lead to changes in the types and levels of both eCBs and OxLs. Since the link between EPA/DHA and eCB/OxL has rarely been studied in hemodialysis patients, the present research endeavor was to explore the relationship between eCB and OxL levels, including those derived from n-3 PUFAs, in hemodialysis patients.

Previous studies have reported elevated levels of uremic toxins and markers of nitric oxide metabolism in subjects suffering from renal dysfunction that were being treated with hemodialysis [26,27,28]. Global metabolite analysis (metabolomics) provides a broad biochemical snapshot that can identify factors associated with the pathophysiology of the progression of disease and disease state [29]. Indeed, by applying the metabolomics in a cross-sectional study, Shah et al. [29] observed differences in the metabolites between the different stages of chronic kidney disease (CKD). Marked changes in arginine metabolism and elevated coagulation and inflammation were found as potential benchmarks of the CKD state. Although metabolomic analysis has been adopted in research on kidney disease patients [30,31,32], to the best of our knowledge, the metabolomics of HDPs at all stages have not been reported. Therefore, the current research is our attempt to fill the knowledge gap for this particular patient population and efficiently document co-occurring metabolic defects in HDPs. Moreover, this secondary effort allowed for the potential identification of novel biomarkers in HDPs and provided an opportunity to integrate any observed alterations in ECS and OxL metabolism to these changes.

The primary hypothesis explored in this preliminary study is that plasma PUFAs are associated with changes in the amounts of AA and long-chain n-3 PUFAs (EPA and DHA), which will reflect increases or decreases in the associated eCB and OxL levels in the blood. Moreover, we hypothesized that differences in the plasma global metabolite profiles between healthy controls and HDPs would highlight the metabolic and physiological consequences of kidney disease and allow a direct evaluation of links between these and the measured eCBs and OxLs. To test these hypotheses, chronic hemodialysis patients (nine female) and age-matched healthy controls (ten female) were recruited. Fatty acid species in the plasma and red blood cells (RBCs), along with an array of plasma eCB and OxL species derived from eighteen- to twenty-two-carbon PUFAs, were measured. In addition, plasma metabolomic analyses were performed. To our knowledge, this is the first study to examine the full array of eCBs and OxLs in hemodialysis patients.

## 2. Results

### 2.1. Fatty Acid Analysis of Plasma and RBCs

The plasma fatty acid and RBC fatty acid composition data are presented in Table 1. Between the healthy controls and the HDPs, the relative abundances of 15:0, 16:0, 20:3n-6, 20:5n-3, and 22:5n-3 were higher in the control group, while the relative levels of 18:0, 20:4n-6, and 22:1n-9 were higher in the female HDP group. The data for the FAME analysis of RBCs did show a lower 20:5n-3 in the HDP group compared to the control group (0.11 ± 0.17 HDPs and 0.49 ± 0.13 controls). The results of higher EPA levels in the plasma and RBCs for controls versus HDPs might have influenced the capacity for the biosynthesis of the related eCBs and OxLs derived from EPA [33], and possibly the downstream actions of these lipid mediators of pain and inflammation [34].

### 2.2. Impact of PMSF on Endocannabinoid (eCB) and Oxylipin (OxL) Data

The addition of PMSF to plasma had subtle impacts on the OxL and eCB results. However, in the presence of PMSF, eight compounds showed subtle decreases (*p* < 0.1) and a reduced variance when the analyses were adjusted for subject as a random effect. These included 2-AG, multiple prostaglandins, and the platelet degranulation-related TXB2, 12-HETE, and 15-HETE. Therefore, the subsequent analyses were conducted using only the PMSF-treated samples.

### 2.3. Endocannabinoids and Oxylipins

A total of 65 eCBs and OxLs were routinely detected in the plasma of study participants, including 15 of 17 (88%) eCBs and 50 of 75 (66%) OxLs. As seen in Figure 1, a PLS-DA did not fully segregate the study population into the control and HDP groups (Q^2^ = 0.30). However, only two members of the HDP group aggregated with the control group, and 29 of the 65 metabolites (45%) had VIP scores > 1. With regards to the eCBs, the HDP group showed higher levels of multiple monoacylglycerols and N-acylethanolamides, but not the canonical endocannabinoids 2-AG or A-EA (Figure 1 and Figure 2A). Of the seven compounds with VIPs < 1, six showed differences by 2-tailed *t*-tests at *p* ≤ 0.05. With regards to the OxLs, the HDP group showed slightly higher levels of two prostanoids, with 6-keto-PGF_1alpha_ different between groups at *p* = 0.1. Of the remaining 18 discriminating metabolites, all showed lower levels, with six showing differences at *p* ≤ 0.05 and eleven reaching *p* ≤ 0.1, including triols, diols, alcohols, epoxides, and hydroperoxides (Figure 1 and Figure 2B). Given the small sample size and the pilot nature of these findings, these results are quite suggestive of differences between the HDP and control groups. The OxL analysis showed that the levels of six compounds were higher in the control females relative to the female HDPs: 11,12-DiHETrE, 14(15)-EpETrE, 14,15-DiHETE, 12-HEPE, 5-HEPE, and 19(20)-EpDPE (Figure 1). The complete results from these analyses, as shown in Table 2, demonstrate lower levels of linoleic acid, α-linolenic acid, oleic acid, and arachidonic acid-derived eCBs, but higher levels of OxLs derived from eicosapentaenoic acid and docosahexaenoic acid in the female controls compared to the female HDPs (Table 2). A complete list of the metabolites, their concentration group means, their VIP scores, and the *p*-values from 2-tailed *t*-tests are included in Appendix A.

The analysis score plot (inset) showing group discrimination and the variable coefficient plot showing metabolite strength in discrimination are displayed. Metabolites with a variable importance in projections (VIP) > 1 were considered to have discriminating power, and are investigated in detail in Figure 2A,B and Table 2. Leave-one-out cross-validation was used to build the model. While showing a tendency towards group discrimination (i.e., Q2 positive), it did not reach full group segregation (Q2 > 0.4). A complete list of metabolites, their concentration group means, their VIP scores, and the *p*-values from 2-tailed *t*-tests are included in Appendix A.

### 2.4. Global Metabolite Profiles

Plasma mass spectrometry-based untargeted metabolomic analysis of HDPs and controls detected 566 unique features, including 192 identified chemical compounds. Of the named metabolites, 82 were found to be different by a *t*-test when compared between the female HDPs and female controls, with 70 higher in female HDPs as presented in Figure 3 and Appendix A. The data were subjected to a partial least squares-discriminant analysis (PLS-DA) to visualize untargeted metabolite differences between the female hemodialysis and healthy female control groups (Figure 2). Upon further analysis, the hierarchical cluster dendrogram was pruned into eight clusters based on the screen plot, and the obtained clusters were used for a multivariate analysis of variance (MANOVA). As shown in Figure 3, the differences between the controls and HDPs revealed specific clusters of the metabolites measured. The data from the PLS were subjected to MANOVA and are presented in Appendix A for ease of interpretation of the metabolites, and for the presentation of means ± SD and the *t*-test *p*-values.

All detected metabolomic features were used for the discriminant analysis, but only metabolites with variable importance in projection scores > 1 were displayed for clarity. The inset score plot demonstrates control and hemodialysis patient (HDP) discrimination, while the loadings plot shows metabolites driving group discrimination. Metabolites with negative and positive loadings in latent variable 1 were lower and higher in the HDP group than in the control group, respectively. A hierarchical cluster analysis of auto-scaled metabolomic data yielded eight unique clusters that were important in group discrimination. The loadings of individual metabolites are color-coded according to hierarchical cluster membership (see the MANOVA analysis in Appendix A), as are their cluster names. Clusters having more than two members with VIPs > 1 are displayed within containing geometric shapes of the same color. The cumulative values for the two factors were: Q^2^ = 0.99; R^2^X = 0.39; and R^2^Y = 0.98. 

### 2.5. Thematic Changes for the Metabolites

To effectively interpret the vast amount of data from the metabolomics analysis, a series of *t*-tests were performed to tease out the interrelationships among hundreds of compounds that are involved in many metabolic pathways. Because of the different representation of the study subjects, the data analyses were grouped based on HDP. Thus, statistical analyses were performed within each grouping, namely between controls and HDPs. The results of these analyses revealed several thematic changes, as illustrated in Figure 3. The principal changes in the metabolomics profiles pointed to an altered amino acid metabolism for essential and nonessential amino acids. This was also reflected by differences in the intermediates of the urea cycle and creatine, which makes them useful as biomarkers of kidney failure.

#### Amino Acids

In the female HDPs, the levels of several amino acids were found to be different compared to those of the female controls. Among these amino acids, leucine, isoleucine, phenylalanine, tryptophan, and valine—essential amino acids—were either higher in the female HDP group, or higher in the female control group (Appendix A). Glutamine, a conditional essential amino acid, was higher, while alanine was lower in the HDPs compared to the controls. A prominent difference was observed in the level of metabolites in the biochemical pathways of amino acids between the HDPs and controls. Only two metabolites in this group were lower in the female HDP group (taurine and 2-hydroxybutanoic acid) compared to the female controls. The data point to a heightened metabolic change in the HDPs compared to the healthy subjects (Appendix A). The levels of creatinine and its catabolic product 1-methylhydantoin were higher in HDPs, echoing the disease state with respect to the diminished kidney function associated with the progression of kidney disease. Other differences were also found between healthy female controls and female HDPs, with higher levels of pyrophosphate and adenosine-5-phosphate in the controls, but higher levels of citrulline and ornithine, two intermediates in the urea cycle, in the HDPs.

## 3. Discussion

### 3.1. Levels of Plasma PUFAs, eCBs, and OxLs in Female Controls and HDPs

Arachidonic acid is the most well-studied PUFA for the biosynthesis of eCBs and OxLs. The two most well-studied ECS ligands, AEA and 2-AG, are arachidonate derivatives. Further, arachidonic acid is the precursor for many immunomodulatory OxLs. In this study, we reported a higher amount of 20:5n-3 (EPA) in the plasma phospholipids of healthy female controls compared to the female HDPs. In contrast, female HDPs had higher arachidonate levels compared to the female controls in their plasma phospholipids. Phospholipid fatty acids are a primary target for evaluation in hemodialysis patients [35]. The differences in the levels of EPA and arachidonate consistently showed changes in both the eCB and OxL amounts between female controls and female HDPs. The observation of low n-3 PUFAs in the plasma of female HDPs is consistent with several previous reports on hemodialysis patients [6,24,25]. Moreover, differences in eCBs are associated with appetite in both male and female hemodialysis patients compared to healthy controls [36].

L-EA, a linoleic acid-derived eCB, has been shown to possess anti-inflammatory effects by inhibiting NF-κB signaling and the expression of pro-inflammatory cytokines (TNF-α, IL-1, and IL-6), as well as inhibiting COX-2 activity and the production of PGE_2_ in mouse RAW264.7 macrophages [37]. In the accompanying in vivo study with a mouse dermatitis model, L-EA successfully lessened 2,4-dinitrofluorobenzene-induced contact dermatitis on ear skin and pro-inflammatory cytokine expression at inflamed sites [37]. It is interesting to note the coexistence of potentially pro-inflammatory arachidonate and the anti-inflammatory L-EA, which were both higher in the female HDPs in this study. These findings might suggest a role for the participation of PUFAs and eCB-like compounds in the complex disease state of hemodialysis and the malfunction of the renal system. L-EA is one of the major N-acylethanolamine species found in peripheral tissues, especially in the intestine, where it is actually the N-acylethanolamine of the highest concentration [38,39]. With respect to the OxLs, arachidonate derivatives represent the well-known and most studied OxLs (eicosanoids), such as prostaglandins, leukotrienes, thromboxanes, and lipoxins. These arachidonate derivatives, especially the prostaglandins and leukotrienes, are generally pro-inflammatory, and their counterparts derived from n-3 PUFAs appear to be less pro-inflammatory [40]. In end-stage renal disease patients on hemodialysis, it has been found that 5-lipoxygenase activity and expression are greatly increased in peripheral blood mononuclear cells, which activates the arachidonate cascade that leads to the formation and release of the reactive oxygen species of pro-inflammatory and pro-atherogenic OxLs and cytokines [41].

Modifying the circulating levels of OxLs by dietary n-3 PUFA supplementation with fish oil improved the condition (measured by proteinuria) of immunoglobulin A nephropathy (IgAN) of the patients and shifted the profile of OxLs towards more EPA and DHA derivatives [23]. An analysis of blood OxLs underscored that the baseline n-3 PUFA status was a significant determinant of OxL status [42]. Grapov et al. examined the linkages between changes in circulating OxLs and eCBs by measuring > 150 plasma lipids in overweight to obese patients with type 2 diabetes and found that these plasma lipidomic profiles reflected the biochemical and physiological changes of this pathological state, independently of obesity [43]. Thus, it is likely that these relationships exist and must be investigated in hemodialysis patients supplemented with n-3 PUFAs. We report for HDPs that, in general, monoacylglycerols were higher, which is consistent with end-stage renal disease patients on maintenance hemodialysis (MHD) [17]. The eCB 2AG did not reach significance (*p* = 0.3; Appendix A). While AEA also appeared higher in the HDP group compared to the control group, with a PLS-DA VIP score of 0.95, this was not significant in a univariate analysis (*p* = 0.40). This is in contrast with the findings of Moradi et al. [18], and the difference between the univariate and multivariate analyses suggests the potential for interactions among the measured metabolites.

### 3.2. HDPs Had Lower Plasma EPA Compared to Controls and Lower 14,15-DiHETE, 12-HEPE, and 5-HEPE

EPA is another major PUFA substrate for a range of eicosanoids that are largely regarded as antagonistic and anti-inflammatory against the actions of those derived from the n-6 PUFA arachidonate with mostly pro-inflammatory activities. The finding that lower levels of EPA and its OxL derivatives were detected in the female HDP group could be an indication that this group of patients may be deficient in their n-3 PUFA status compared to the healthy controls. Indeed, it has been reported that dietary EPA supplementation is associated with an enhanced in vivo production of EPA-derived OxLs, e.g., 5-HEPE and 15-HEPE, as shown in 116 human subjects (68% female, 20–59 years old) given a fish oil supplement (2 g EPA + 1 g DHA, daily) in a randomized placebo-controlled study [44]. Supplementing with n-3 PUFAs increased the n-3 OxL levels from 2- to 5-fold and reduced those of n-6 origin by approximately 20% in healthy volunteers during 4 weeks of treatment with prescription n-3 PUFA ethyl esters (4 g/day) [45]. This supports our premise that n-3 PUFAs should be incorporated into the treatment regimen of hemodialysis patients [24,25]. Furthermore, the higher amounts of arachidonic acid-derived epoxy fatty acids in the healthy female controls compared to the female HDPs must be evaluated in future studies.

It is well-known that unsaturated fatty acids are often converted to epoxides by P450 monooxygenases [46,47,48,49]. Numerous studies have shown that these cytochrome P450-mediated arachidonate metabolites could play multiple physiological and pathological roles on the cerebral microvasculature to mediate or alleviate the pathogenesis of cerebrovascular diseases [50,51]. HDPs are known to be in an inflammatory state [3]. In support of the heightened state of inflammation, the pro-inflammatory cytokine IL-6 concentration is a robust predictor of death in hemodialysis patients [52]. Noori et al. corroborated this finding, reporting that higher serum albumin, prealbumin, and creatinine were associated with greater survival, whereas C-reactive protein and IL-6 were associated with increased mortality in hemodialysis patients [53]. Further supporting the link between the inflammatory state and kidney disease, Na et al. showed that a significant association exists between the WBC count and the risk for developing CKD in women, but not in men [54]. EETs are regulated at multiple levels, from production to epoxide hydrolase-dependent degradation [49,55,56]. They are also integrally linked to the renin–angiotensin system and blood pressure control.

### 3.3. Amino Acid and Metabolite Differences between Healthy Controls and HDPs

Proteins and amino acids are important nutrients for maintaining muscle mass and biosynthetic capacity in the body. Plasma amino acid levels are believed to directly reflect disturbances in protein and amino acid metabolism and their inter-organ exchanges in renal dysfunction, since compromised renal function disrupts metabolism and as a consequence, it alters circulating metabolite profiles [57,58]. Qi et al. have reported significant changes in endogenous metabolites, including glycolysis products (glucose and lactate), amino acids (valine, alanine, glutamate, and glycine), and organic osmolytes (betaine, myo-inositol, taurine, and glycerophosphocholine) in different stages of CKD in a pilot metabolic profiling study that compared 80 patients in four stages of CKD and 28 healthy controls [59]. In patients suffering from acute kidney injury, Sun et al. also showed increased serum acylcarnitines and amino acids (methionine, homocysteine, pyroglutamate, asymmetric dimethylarginine, and phenylalanine), deceased arginine, and several decreased lysophosphatidylcholines compared to healthy controls [60]. Among these, increases in homocysteine, asymmetric dimethylarginine, and pyroglutamate have been shown as biomarkers of cardiovascular and renal diseases, while augmented acylcarnitines could serve as biomarkers of defective oxidative fatty acid metabolism [60,61]. In the current study, the levels of several amino acids were found to be different between the female HDPs and the female controls, with two (phenylalanine and glutamine) higher in the HDPs, while another two (tryptophan and alanine) were lower in the HDPs.

It has been reported that free amino acids, either from dietary intake or from protein catabolism, are substantially retained in the plasma of patients with end-stage renal disease, where dialysis is adopted to correct the imbalance [57]. In our study, we found a similar phenomenon, where the major differences in amino acid metabolites were higher in HDPs compared to controls. It was clearly shown that there was a distinct difference in amino acid metabolism and the levels of their metabolites between the HDPs and the healthy female controls. These changes might be related to large differences in the systemic use of amino acids and intermediary metabolism. Recently, the upregulation of the CB2 receptor was found to play a primary role in the mitochondrial dysfunction of renal tubular cells [62]. Thus, understanding the full extent of differences in eCBs and OxLs in the present study helps identify the relationship between dietary PUFAs and these bioactive lipids in the HDPs.

The creatinine concentration rises when the kidney function is compromised. Therefore, blood creatinine alone or in conjunction with the 24 h urine creatinine level is traditionally used as a clinical biomarker for assessing kidney function [63]. In our investigation, the female HDPs had higher plasma levels of creatinine (6-fold increase) compared to the female controls, which clearly shows a distinct difference between patients with kidney disease who rely on hemodialysis and those without compromised renal function. However, creatinine concentrations in the blood could be affected by age, gender, muscle mass, medication, or hydration status [64]. In the end-product metabolism of amino acids, we found that the HDP group’s urea levels were twice that of controls, and two intermediates of the urea cycle, namely citrulline and ornithine, were elevated to the same extent. The findings of Shah et al. [29] are consistent with these results.

In a recent article [65], plasma metabolites were measured in 19 MHD patients that were receiving end-stage renal dialysis and 12 healthy controls to identify metabolites unique to the dialysis patients. The subjects and patients consisted of both males and females. The analysis revealed that 30 metabolites were higher and 33 lower in the MHD patients compared to the controls in the study; a few of these were observed in our current investigation. Here, in the plasma, we observed higher levels of hippuric acid and several indole-derivative compounds, and lower levels of lactic acid, in the MHD patients compared to the controls, which is consistent with the findings of Chen et al. [65]. The study by Chen and colleagues [65] did not include any eCBs or OxLs. Nonetheless, with the number of subjects in our study, some metabolite levels were similar, and our findings for OxLs emphasize the differences in the inflammatory state for HDPs.

### 3.4. Implications, Strengths, and Limitations

eCBs and OxLs are changed in HDPs compared to control women. These findings would implicate a possible link for OxLs and the ECS as candidate targets for chronic kidney disease due to their role in inflammation. The study limitation is the small sample size; however, many of our results for eCBs and metabolites are consistent with other investigators. The dialysis patients had a high BMI, as did the controls, and they were matched by age, height, and weight with the controls. Although the subjects had a high BMI and appeared to have adapted to the hemodialysis, our data might not be the same for patients with a lower-range BMI. Yet, both the controls and the HDP group had similar BMI values. The HDP group had a higher incidence of tobacco use (four of nine participants), which may have influenced some results. However, recent reports indicate few impacts on OxL levels [66]. While tobacco smoke exposure may influence the endocannabinoid system via nicotine exposure [67], to the best of our knowledge, its influences on circulating eCBs have not been reported.

## 4. Materials and Methods

### 4.1. Subjects and Design

Study subjects were recruited at Indiana University School of Medicine, Indianapolis, IN and included 9 female HDPs (hemodialysis patients) and 10 age-, BMI-, and diabetes status-matched non-HDP female controls (Table 3). All subjects were either overweight or obese, with a BMI of 31.0 ± 9.3 in the patients and 31.1 ± 5.4 in the control group. Furthermore, the differences in ethanol use and smoking between the groups could be a limitation. The average time on dialysis was 10.0 ± 11 years for the female patients. The study cohort was derived from our previous study [36]. The 9 HDPs were part of a larger cohort of 20 dialysis patients of mixed sex. After identifying this female cohort, we then recruited healthy female volunteers that were matched to our cohort using frequency matching by age. SNAQ appetite scores were obtained for all subjects, with values of 12.00 ± 3.39 for the female patients and 14.90 ± 2.60 for the control females [36]. Blood samples were obtained immediately pre-dialysis from arteriovenous blood sampling from the dialysis tubing for all HDPs, and from the cannulation of blood veins in the controls. Blood samples for eCBs and OxLs were treated with the protease inhibitor phenylmethanesulfonylfluoride (PMSF), a fatty acid amide hydrolase (FAAH) inhibitor added to a set of samples to prevent the breakdown of anandamide (i.e., arachidonoyl ethanolamide (AEA)) [36]. The samples for the FAME analysis of plasma and RBCs were not treated as such. This was a secondary analysis of an exploratory study to assess the differences between cohorts, and as such, no power calculations were needed. The severity of the patients’ kidney function was uniform among the hemodialysis population because they were all on dialysis, which indicates minimal to no residual kidney function remaining. No markers of inflammation were measured in these patients. The study was initiated after written informed consent from the study participants and approved by the Institutional Review Board at the University of Indiana, School of Medicine (ClinicalTrials.gov Identifier: NCT01477515).

### 4.2. Fatty Acid Analysis of Plasma and RBCs

Human plasma and RBC samples were processed to determine the fatty acid composition. The samples processed were from all 19 subjects. Briefly, 200 µL of RBCs or 100 µL of plasma were extracted with chloroform/methanol (2:1, vol/vol). Lipids from the plasma were further separated into polar and neutral fractions by solid-phase extraction using silica cartridges (300 mg filling, Alltech) after eluting the neutral fraction with chloroform and the polar fraction with methanol [68]. Only the polar lipids were used for further analysis, based on several fatty acid analyses of HDPs. The resulting extracted lipids were treated with 0.5 N NaOH in methanol, and fatty acid methyl esters (FAME) were prepared by esterification using boron trifluoride (BF_3_) in methanol (10% *w/w*, Supelco Inc. Bellefonte, PA, USA). The FAME were concentrated in isooctane (HPLC grade, Fisher Scientific, Pittsburg, PA, USA) and analyzed by gas chromatography (GC) (HP 7890A series, autosampler 7693, GC ChemStation Rev.B.04.03, Agilent Technologies, Palo Alto, CA, USA) with a DB-225 column (30 m, 0.25 mm i.d., 0.15 mm film thickness, Agilent Technologies, Palo Alto, CA, USA) equipped with a flame ionization detector [68]. Sample peaks were identified by comparison to authentic FAME standards (Nu-Chek-Prep Inc., Elysian, MN, USA). The results of the FAME analysis were obtained by area percentage reports.

### 4.3. Measurement of Endocannabinoids (eCBs) and Oxylipins (OxLs)

Measurements of eCBs and OxLs were performed by LC-MS/MS of 250 µL of plasma, thawed on ice, placed into solid-phase extraction column cartridges on a vacuum manifold, spiked with deuterated eCB/OxL internal standards, diluted to 20% MeOH/0.1% acetic acid, and gravity-loaded onto a 60 mg Oasis-HLB solid-phase extraction column, followed by vacuum-drying with ambient air. The columns were then wetted with 0.2 mL MeOH and eluted with 0.5 mL ethyl acetate by gravity. The solvent was removed by vacuum, the samples were reconstituted in 50 µL MeOH containing internal standards, samples were filtered at 0.1 µm, and they were then analyzed by UPLC-(ESI)MS/MS by back-to-back (+)-mode/(−)-mode injections for 17 eCBs and eCB-like compounds and 58 OxL profiles, respectively, in the Newman Laboratory, as previously reported [42,43].

### 4.4. Measurement of Metabolites

Human plasma samples were analyzed for global metabolites by following the method of Fiehn et al. [69]. An Agilent 6890 gas chromatograph, controlled using Leco ChromaTOF software, and a 30 m × 0.25 mm i.d. × 0.25 µm 95% dimethyl/5% diphenyl polysiloxane film Rtx-5Sil MS column with an additional 10 m integrated guard column were used. Pure helium (99.9999%) with a built-in purifier was used at a constant flow of 1 mL per min. The oven temperature was held constant at 50 °C for 1 min, and then ramped up by 20 °C per min to 330 °C and held constant for 5 min. The mass spectrometry instrument was a Leco Pegasus IV time-off light mass spectrometer, controlled using Leco ChromaTOF software, version 2.32. For the sample introduction, the transfer line temperature between the gas chromatograph and the mass spectrometer was set to 280 °C; electron impact ionization at 70 V was employed, with an ion-source temperature of 250 °C. Data acquisition was performed after a 290 s solvent delay, filament 1 was turned on, and the mass spectra were acquired at the mass resolving power R = 600 from *m/z* 85–500, at 20 spectra per second and a 1550 V detector voltage, without turning on the mass defect option. Recording ended after 1200 s. The instrument performed auto-tuning for mass calibration using FC43 (perfluorotributylamine) before starting the analysis sequences. The results of the eCB and OxL analysis of plasma samples (with and without PMSF) from 2 of the same subjects were similar [36].

### 4.5. Statistical Analyses

Data for plasma fatty acids, eCBs, OxLs, and metabolites were expressed as the means and standard deviations. Group differences in fatty acid analyses were evaluated by the *t*-test procedure using SAS (v 9.3; SAS Institute, Inc., Cary, NC, USA). Analyses of eCBs, OxLs, and metabolomics were conducted in JMP Pro (v 16.1; SAS Institute, Inc., Cary, NC, USA). Values were log-transformed and normality was evaluated prior to the analyses. The impact of PMSF on eCB and OxL measurements was assessed by least-squares mean regressions, with the subject as a random effect. Partial least squares-discriminate analysis (PLS-DA) was used to determine if a multivariate metabolite analysis could segregate experimental groups. The calculated model quality metrics included Q^2^, R^2^X, and R^2^Y, with Q^2^ > 0.4 indicating full group segregation. Metabolites with variable importance in projection (VIP) scores > 1 were considered significant discriminating variables. Group mean differences were assessed by 2-tailed *t*-tests, with and without an adjustment for multiple comparisons, using the Benjamini–Hochberg false discovery rate correction procedure at q = 0.2 [70]. Based on previous studies for the OxLs and eCBs, a power analysis revealed that an n value of 8–10 allowed for discrimination of these patients [36].

Metabolomic analyses used only the PMSF-containing sample set. Metabolomic data dimensionality was reduced by grouping highly correlated variables using hierarchical cluster analysis, as described by Ward [71]. Clusters were assigned using auto-scaled data to ensure that all metabolites were equally considered in the statistical models [72]. The hierarchical cluster dendrogram was pruned into 8 clusters based on the screen plot, and the obtained clusters were used for multivariate analysis of variance (MANOVA). MANOVA analysis was performed using hierarchical clusters, with the metabolite as a fixed effect and the subject as a random effect. Prior to analysis, data were normalized using the Johnson transformation. Clustering was performed separately for the targeted and untargeted analyses. MANOVAs and *t*-tests were used to evaluate group differences in metabolite abundances. Multiple comparison corrections of MANOVA and *t*-test *p*-values used the Benjamini–Hochberg false discovery rate correction at q = 0.2 [70]. A PLS-DA was used to visualize untargeted metabolite differences between the HDP and control groups. Analyses were performed using auto-scaled data after the imputation of metabolites missing was detected in > 70% of study participants. All metabolites were used to perform the analysis; however, only metabolites with VIP > 1 are displayed in figures for clarity of presentation. VIP scores were negatively correlated with the *t*-test *p*-values.

## 5. Conclusions

In summary, female dialysis patients have alterations in eCB and OxL species that are associated with plasma PUFAs, following our hypothesis. This important diet relationship should be further investigated to ascertain if these lipid biomarkers are associated with or help mediate disease conditions. Furthermore, in our study we found that plasma phospholipid EPA levels (and RBCs) were higher in the healthy control group compared to the HDP group. While the majority of OxLs also followed this pattern, the eCB-like compounds measured were generally higher in the HDP group. Thus, our results suggest that OxLs and the ECS are potential targets for altering the course of chronic kidney disease [73]. In addition, amino acid metabolism is significantly altered by the kidney disease state. Our investigation is the first to report a comprehensive examination of the status of eCBs, OxLs, and other metabolites in female control and female hemodialysis patients. Combined with our previous work on the benefits on long-chain n-3 PUFAs in this population, and the emphasis on the DHA and EPA status in chronic kidney disease [74], the current research, although preliminary, identifies new opportunities for evaluating the role of dietary PUFAs and eCBs/OxLs in clinical disease for kidney dialysis patients. Given the small number of subjects used in this study as a limitation (albeit the fact that we reported some similar findings to other investigators for eCBs and systemic metabolites), validation is needed in a larger cohort.

## Figures and Tables

**Figure 1 ijms-23-09781-f001:**
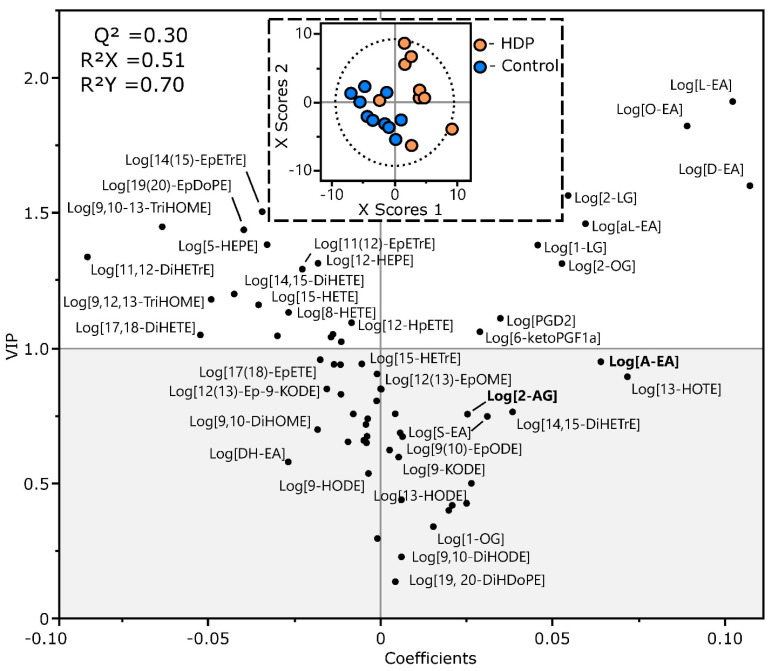
Partial least squares-discriminate analysis of control and hemodialysis patient (HDP) endocannabinoids and oxylipins.

**Figure 2 ijms-23-09781-f002:**
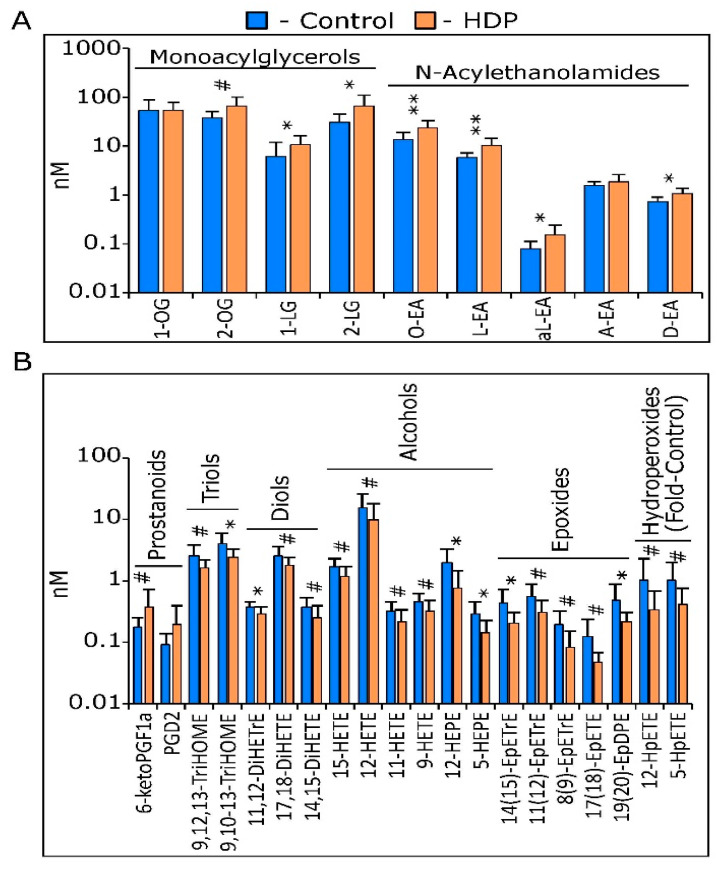
Selected plasma lipids in controls (*n* = 10) and hemodialysis patients (HDP; *n* = 9), measured by UPLC-tandem mass spectrometry. With the exception of 1-OG and AEA, all metabolites showed group discrimination power in a partial least squares-discriminate analysis with variable importance in projection scores > 1 (Figure 1). (**A**) Endocannabinoid and endocannabinoid-like compounds, including monoacylglycerols and N-acylethanolamides. (**B**) Oxylipins, including prostanoids, triols, diols, alcohols, epoxides, and hydroperoxides. Hydroperoxide levels were estimated from internal standard corrected-area responses and are expressed as fold-control. Results are expressed as the mean ± SD. The *p*-values < 0.05 are rounded to 1 significant figure. Group mean differences were assessed on log-transformed data by 2-tailed *t*-tests, with annotations indicating *p*-values (** ≤ 0.01; * ≤ 0.050, # ≤ 0. 10).

**Figure 3 ijms-23-09781-f003:**
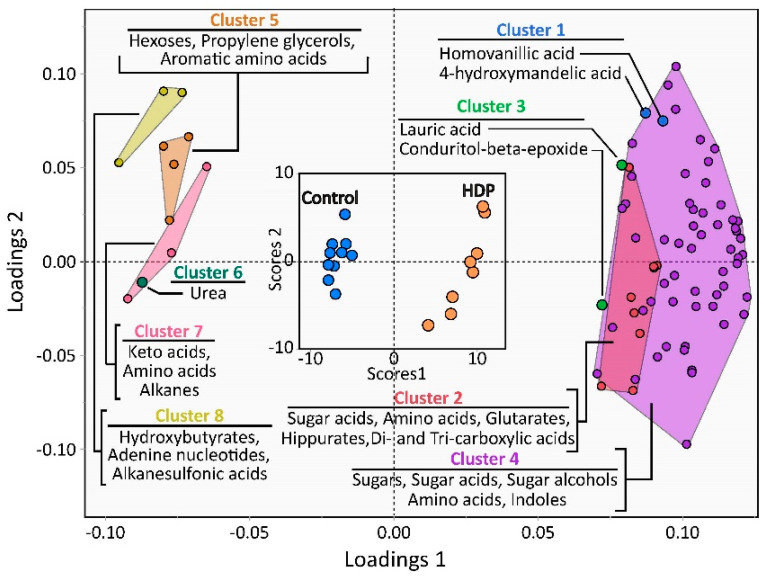
Partial least squares-discriminant analysis of untargeted metabolomic data.

**Table 1 ijms-23-09781-t001:** Plasma fatty acid percent composition of polar lipids and RBCs in female healthy control subjects and female hemodialysis patients.

Fatty Acids	Plasma Polar Lipid Fatty Acids			RBC Fatty Acids		
	Controls (*n* = 10)	HDPs (*n* = 9)	*p*-Values	Controls (*n* = 10)	HDPs (*n* = 9)	*p*-Values
14:0	1.23 ± 0.76	1.28 ± 0.63	0.9	0.17 ± 0.18	0.06 ± 0.13	0.14
15:0	0.09 ± 0.07	ND	-	ND	ND	-
16:0	24.15 ± 1.34	21.86 ± 1.44	0.0022	18.71 ± 1.17	18.37 ± 1.14	0.83
t16:1n-7	0.15 ± 0.06	0.10 ± 0.09	0.2	0.03 ± 0.02	0.04 ± 0.03	0.53
16:1n-7	0.67 ± 0.36	0.40 ± 0.26	0.08	0.03 ± 0.05	ND	-
17:0	0.34 ± 0.05	0.33 ± 0.04	0.6	0.32 ± 0.26	0.06 ± 0.12	0.014
18:0	18.59 ± 1.45	20.47 ± 1.61	0.016	12.95 ± 1.27	12.91 ± 0.88	0.50
18:1n-9	10.82 ± 1.56	10.95 ± 2.23	0.9	16.78 ± 1.79	15.69 ± 1.85	0.46
18:1n-7	1.42 ± 0.23	1.66 ± 0.28	0.06	1.32 ± 0.17	1.47 ± 0.19	0.05
18:2n-6	18.16 ± 3.29	18.40 ± 2.28	0.9	11.25 ± 1.48	11.31 ± 1.73	0.93
18:3n-6	0.15 ± 0.11	0.05 ± 0.08	0.042	0.03 ± 0.06	ND	-
18:3n-3	0.34 ± 0.12	0.35 ± 0.10	0.8	0.04 ± 0.07	ND	-
20:0	0.16 ± 0.02	0.18 ± 0.04	0.2	0.04 ± 0.06	0.06 ± 0.13	0.71
20:1n-9	0.15 ± 0.06	0.19 ± 0.03	0.05	0.09 ± 0.11	0.18 ± 0.28	0.36
20:2n-6	0.34 ± 0.07	0.32 ± 0.06	0.5	0.26 ± 0.10	0.19 ± 0.19	0.34
20:3n-6	3.14 ± 0.82	2.15 ± 0.81	0.017	1.74 ± 0.28	1.27 ± 0.45	0.013
20:4n-6	11.47 ± 1.71	13.61 ± 1.87	0.019	15.98 ± 1.40	15.71 ± 3.49	0.83
20:5n-3	0.71 ± 0.28	0.38 ± 0.09	0.005	0.49 ± 0.13	0.11 ± 0.17	0.001
22:0	0.08 ± 0.06	0.04 ± 0.08	0.2	0.33 ± 0.12	0.41 ± 0.33	0.47
22:1n-9	0.18 ± 0.10	0.31 ± 0.14	0.032	ND	0.03 ± 0.08	-
22:4n-6	0.55 ± 0.15	0.52 ± 0.06	0.5	3.73 ± 0.71	4.04 ± 0.95	0.44
22:5n-6	0.43 ± 0.21	0.35 ± 0.10	0.3	0.84 ± 0.23	0.68 ± 0.30	0.22
22:5n-3	0.88 ± 0.11	0.71 ± 0.08	0.0013	1.98 ± 0.28	1.77 ± 0.47	0.25
22:6n-3	3.11 ± 1.57	2.68 ± 0.78	0.5	4.06 ± 1.40	3.63 ± 1.45	0.52
24:0	0.06 ± 0.05	0.03 ± 0.07	0.4	0.95 ± 0.15	0.93 ± 0.73	0.94
24:1n-9	0.09 ± 0.08	0.13 ± 0.08	0.3	0.91 ± 0.14	1.10 ± 0.90	0.54
Total SFA	44.71 ± 1.33	44.19 ± 2.13	0.5	36.11 ± 0.86	36.83 ± 3.87	0.54
Total MUFA	13.33 ± 1.77	13.65 ± 2.52	0.8	15.76 ± 1.30	16.32 ± 2.05	0.38
Total PUFA	39.29 ± 1.83	39.51 ± 1.87	0.8	40.39 ± 1.45	38.90 ± 7.83	0.50
Total n-6 PUFA	34.25 ± 1.88	35.40 ± 1.74	0.2	33.82 ± 1.93	33.19 ± 5.43	0.75
Total n-3 PUFA	5.04 ± 1.66	4.11 ± 0.91	0.15	6.56 ± 1.29	5.51 ± 1.93	0.18
LC n-6 PUFA	15.94 ± 2.13	16.94 ± 1.58	0.26	22.5 ± 0.93	21.88 ± 4.34	0.68
LC n-3 PUFA	4.71 ± 1.67	3.77 ± 0.85	0.15	6.52 ± 1.30	5.51 ± 1.93	0.19
LC n-6/n-3 Ratio	3.72 ± 1.29	4.65 ± 0.86	0.087	3.61 ± 0.88	4.79 ± 2.73	0.24

The fatty acid data were measured as fatty acid methyl esters as area % means ± SD of plasma polar lipid fatty acids and RBCs. Authentic external standards were used for peak identification of chromatogram data during integration. The sensitivity of the GC detector is at 10 ng/peak. HDPs = hemodialysis patients; LC n-6 = 20:2n-6 + 20:3n-6 + 20:4n-6 + 22:4n-6 + 22:5n-6; LC n-3 = 20:5n-3 + 22:5n-3 + 22:6n-3. ND = not detected.

**Table 2 ijms-23-09781-t002:** Plasma endocannabinoid and oxylipin subset levels in healthy controls and hemodialysis patients ^*a*,*b*^.

Compound	Parent Fatty Acid	Units	Control	HDP	VIP	*p*-Value
*Monoacylglycerols*						
2-OG	OA	nM	38.3 ± 13	65.8 ± 38	1.3	0.1
1-LG	LA	nM	6.25 ± 6	10.6 ± 5.9	1.4	0.033
2-LG	““	nM	30.5 ± 15	67 ± 42	1.6	0.026
*N-Acylethanolamides*						
O-EA	OA	nM	14.1 ± 5.1	24.5 ± 9.7	1.8	0.0086
L-EA	LA	nM	5.8 ± 1.5	10.4 ± 4.1	1.9	0.0017 *^b^*
αL-EA	αLEA	nM	0.082 ± 0.03	0.154 ± 0.091	1.5	0.024
A-EA	AA	nM	1.59 ± 0.3	1.88 ± 0.71	1.0	0.5
D-EA	AdA	nM	0.734 ± 0.19	1.06 ± 0.32	1.6	0.023
*Prostanoids*						
6-ketoPGF1a	AA	nM	0.177 ± 0.078	0.377 ± 0.34	1.1	0.1
PGD2	““		0.0938 ± 0.046	0.196 ± 0.2	1.1	0.2
*Triols*						
9,12,13-TriHOME	LA	nM	2.51 ± 1.3	1.59 ± 0.6	1.2	0.063
9,10-13-TriHOME	““	nM	3.9 ± 1.9	2.35 ± 0.89	1.4	0.019
*Alcohols*						
15-HETE	AA	nM	1.66 ± 0.68	1.19 ± 0.51	1.1	0.1
12-HETE	““	nM	15.6 ± 10	9.68 ± 8.4	1.0	0.1
11-HETE	““	nM	0.318 ± 0.13	0.221 ± 0.12	1.0	0.1
9-HETE	““	nM	0.46 ± 0.15	0.316 ± 0.17	1.2	0.070
12-HEPE	EPA	nM	1.92 ± 1.4	0.76 ± 0.67	1.3	0.037
5-HEPE	““	nM	0.295 ± 0.16	0.147 ± 0.077	1.4	0.026
*Hydroperoxides*						
12-HpETE	AA	Fold-C	1.0 ± 1.3	0.35 ± 0.35	1.1	0.087
5-HpETE	AA	Fold C	1.0 ± 1.0	0.42 ± 0.33	1.1	0.1
*Epoxides*						
14(15)-EpETrE	AA	nM	0.437 ± 0.28	0.203 ± 0.11	1.5	0.014
11(12)-EpETrE	““	nM	0.551 ± 0.32	0.309 ± 0.17	1.3	0.06
8(9)-EpETrE	““	nM	0.195 ± 0.13	0.0822 ± 0.068	1.0	0.1
17(18)-EpETE	EPA	nM	0.124 ± 0.11	0.0478 ± 0.019	1.0	0.1
19(20)-EpDPE	DHA	nM	0.484 ± 0.38	0.212 ± 0.09	1.4	0.019
*Diols*						
11,12-DiHETrE	AA	nM	0.372 ± 0.084	0.287 ± 0.089	1.3	0.031
17,18-DiHETE	EPA	nM	2.5 ± 1.1	1.79 ± 0.66	1.0	0.1
14,15-DiHETE	““	nM	0.378 ± 0.16	0.247 ± 0.15	1.2	0.057

*a*—all values are mean ± SD. Variable importance in projection (VIP) scores are reported from partial least squares-discriminant models of all measured metabolites. The *p*-values are 2-tailed *t*-tests comparing group means. *b*—difference in L-EA between groups that survived false discover rate correction at q = 0.2. Abbreviations: AA = arachidonic acid; AdA = adrenic acid; αLEA = alpha-linoleic acid; DHA = docosahexaenoic acid; EPA = eicosapentaenoic acid; Fold-C = fold-control, i.e., control ÷ HDP; HDP = hemodialysis patient; LA = linoleic acid; OA = oleic acid.

**Table 3 ijms-23-09781-t003:** Characteristics of female control subjects and female hemodialysis patients.

Characteristics	Units	Controls (*n* = 10)	Hemodialysis Patients (*n* = 9)
Diabetes mellitus	# positive	2	2
NSAID	# positive	4	5
Age	yr	54.82 ± 4.99	59.31 ± 12.81
Medical review			
Time on dialysis	yr	0	10.00 ± 10.52
Age	yr	54.43 ± 4.90	58.02 ± 15.06
Weight	kg	84.01 ± 11.5	82.71 ± 23.68
Height	m	1.65 ± 0.08	1.63 ± 0.06
BMI	kg/m^2^	31.08 ± 5.43	31.03 ± 9.26
3 mo. weight change	kg	No value	0.84 ± 1.82
6 mo. weight change	kg	No value	0.67 ± 1.26
Tobacco	# positive	0	4
ETOH	# positive	5	1
Marijuana	# positive	0	0

Values are means ± SD. The # values for diabetes mellitus and NSAID are the number of cases.

## Data Availability

The only available data can be found in the Appendix A.

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
