# Peer review of "Blood Levels of Endocannabinoids, Oxylipins, and Metabolites Are Altered in Hemodialysis Patients"

_ijms, 2022, doi:10.3390/ijms23179781_

Round 1

Reviewer 1 Report

In this clearly and well written article Authors describe and analyze in depth in a pilot study blood levels of endocanabinoids and oxylipins by chromatography and Mass Spectrometry in 9 hemodialysis overweight women compared with 10 overweight healthy women. Authors found significant anomalies which might be relevant for dialysis-associated inflammation and cardiovascular burden of ESKD.
I have only 3 minor comments:

First: Line 378 Authors stated that the patients " were on early stage of hemodialysis "(usually defined as the first 6 months after initiation of dialysis) whereas line 381-382 they wrote that "the time on dialysis was 10+/-11 years" ; thus I think that we cannot consider these patients on the early stage of hemodialysis according to the accepted definition.

Second : Line 388 : I think there is an error : Authors stated that blood samples were taken "from the dialysis tubing for all the 19 subjects " ; in fact for the 9 dialyzed patients but not for the 10 healthy women (normal cannulation of blood veins,  I guess).

Third : Finally, I think that authors should discuss in the limitations section one important fact : they studied overweight dialyzed patients who had a good survival with hemodialysis(>10 years ; a good example of the paradox of better survival of obese patients in dialysis): we might hypothesize that hemodialysis patients with normal weight and frail dialysis patients with worst dialysis survival can have different pattern and levels of endocanabinoids and oxylipins owing to their more severe inflammatory state , even malnutrition associated with frailty, and higher risk of cardiovascular events.

Author Response

I have only 3 minor comments:

First: Line 378 Authors stated that the patients " were on early stage of hemodialysis "(usually defined as the first 6 months after initiation of dialysis) whereas line 381-382 they wrote that "the time on dialysis was 10+/-11 years"; thus I think that we cannot consider these patients on the early stage of hemodialysis according to the accepted definition.

Response:  Regarding lines 378, and 381-382, yes, the mean value is very large for time is large.  We now removed early-stage hemodialysis from the manuscript.

Second : Line 388 : I think there is an error : Authors stated that blood samples were taken "from the dialysis tubing for all the 19 subjects " ; in fact for the 9 dialyzed patients but not for the 10 healthy women (normal cannulation of blood veins,  I guess). 

Response:  Thank you for the comments.  Blood from controls was by cannulation of blood veins. This is added at line 399.  The sentence on line 383 is referring to the larger cohort of 20 dialysis patients of mixed sex that the subjects for the current study was taken from.  This number 20 was removed from the methods and materials to prevent confusion.

Third : Finally, I think that authors should discuss in the limitations section one important fact : they studied overweight dialyzed patients who had a good survival with hemodialysis(>10 years ; a good example of the paradox of better survival of obese patients in dialysis): we might hypothesize that hemodialysis patients with normal weight and frail dialysis patients with worst dialysis survival can have different pattern and levels of endocanabinoids and oxylipins owing to their more severe inflammatory state , even malnutrition associated with frailty, and higher risk of cardiovascular events.

Response: Thanks for the comment. As suggested, we included new text starting on line 376 to address this concern about BMI of hemodialysis patients.

Reviewer 2 Report

The objective of author’s investigation was to examine levels of endocannabinoids (eCB) and oxylipins (OxL) in female hemodialysis patients (HDP ) compared to healthy matched female controls. The main proposal of the present research was that in hemodialysis patients changes in eCB and OxL levels correlates with differences in corresponding polyunsaturated fatty acids (PUFA). 

The authors summarize that “metabolic differences in eCB and OxL help explain the severe inflammatory and cardiovascular disease manifested by HDP should be explored in future studies”. So, it is very important to correlate measured difference with current inflammation status of studied patients. 

Main remark is that authors did not provide any information on patient’s inflammation status based on standard tests (e.g. C-reactive protein, IL-6, etc.) and severity of kidney disfunction. This information authors may provide in Supplement section. These data are essential for characterization of homogeneity or heterogeneity of patient’s cohort. 

At the end of discussion authors mentioned that within HDP-group there were patients that “had a higher incidence of tobacco use (4 of 9 participants), which may influence some results” but they did not make attempt to extract data on tobacco group and compare them with result of whole cohort. This simple operation can support or not the above-mentioned remark that tobacco smokers had a different metabolic profile than that of non-smokers. 

Minor remark. The caption to figure 3 should be made more descriptive, especially: with regard to the insert.

Author Response

Comments and Suggestions for Authors

The objective of author’s investigation was to examine levels of endocannabinoids (eCB) and oxylipins (OxL) in female hemodialysis patients (HDP ) compared to healthy matched female controls. The main proposal of the present research was that in hemodialysis patients changes in eCB and OxL levels correlates with differences in corresponding polyunsaturated fatty acids (PUFA). 

The authors summarize that “metabolic differences in eCB and OxL help explain the severe inflammatory and cardiovascular disease manifested by HDP should be explored in future studies”. So, it is very important to correlate measured difference with current inflammation status of studied patients. 

Response: Thanks for the comment.

Main remark is that authors did not provide any information on patient’s inflammation status based on standard tests (e.g. C-reactive protein, IL-6, etc.) and severity of kidney disfunction. This information authors may provide in Supplement section. These data are essential for characterization of homogeneity or heterogeneity of patient’s cohort. 

Response: Thanks for the comment. The best we can do at this time is point out the general condition of the hemodialysis population.  For example, the severity of the patients’ kidney function was uniform among the hemodialysis population because they were all on dialysis, which indicates minimal to no residual kidney function remaining.  No markers of inflammation were measured in these patients.  This additional information is now added to the manuscript starting on line 404.

At the end of discussion authors mentioned that within HDP-group there were patients that “had a higher incidence of tobacco use (4 of 9 participants), which may influence some results” but they did not make attempt to extract data on tobacco group and compare them with result of whole cohort. This simple operation can support or not the above-mentioned remark that tobacco smokers had a different metabolic profile than that of non-smokers. 

Response: Thanks for the comments.  Our results did not show an effect of smoking on the measurements and the number of tobacco use was 4 out of nine subjects. We do not have information on the amount of smoking.  Further, without additional subjects it is rather difficult to interrogate the data on the reported smoking use in this group.  This would be a good area of pursuit in future studies.

Minor remark. The caption to figure 3 should be made more descriptive, especially: with regard to the insert.

Response: We did revise Figure 3 based on your comment with appreciation.  We believe the new figure 3 is much improved for presentation of the findings.